

**Effects of grain size and seawater salinity on magnesium hydroxide**
**dissolution and secondary calcium carbonate precipitation kinetics:**
**implications for ocean alkalinity enhancement**
Charly A. Moras[1*], Tyler Cyronak[2], Lennart T. Bach[3], Renaud Joannes-Boyau[1] and Kai G. Schulz[1]
[1]Faculty of Science and Engineering, Southern Cross University, Lismore, NSW, Australia
[2]Institute for Coastal Plain Science, Georgia Southern University, Savannah, GA, USA
[3]Ecology & Biodiversity, Institute for Marine and Antarctic Studies, University of Tasmania, Hobart, TAS, Australia
*Correspondence:* Charly A. Moras (c.moras.10@student.scu.edu.au)



**Abstract.** Understanding the impact that mineral grain size and seawater salinity have on magnesium hydroxide ($Mg(OH)_2$)
dissolution and secondary calcium carbonate ($CaCO_3$) precipitation is critical for the success of ocean alkalinity enhancement.
We tested the $Mg(OH)_2$ dissolution kinetics in seawater using three $Mg(OH)_2$ grain sizes (<63, 63-180 and >180 µm) and at
three salinities (~36, ~28 and ~20). While $Mg(OH)_2$ dissolution occurred quicker the smaller the grain size, salinity did not
significantly impact measured rates. Our results also demonstrate that grain size can impact secondary $CaCO_3$ precipitation,
suggesting that an optimum grain size exists for ocean alkalinity enhancement (OAE) using solid $Mg(OH)_2$. Of the three grain
sizes tested, the medium grain size (63-180 µm) was optimal in terms of delaying secondary $CaCO_3$ precipitation. We
hypothesize that in the lowest grain size experiments, the higher surface area provided numerous $CaCO_3$ precipitation nuclei,
while the slower dissolution of bigger grain size maintained a higher alkalinity/pH at the surface of particles, increasing $CaCO_3$
precipitation rates and making it observable much quicker than for the intermediate grain size. Salinity also played a role in
$CaCO_3$ precipitation where the decrease in magnesium (Mg) allowed for secondary precipitation to occur more quickly, similar
in effect size to another known inhibitor, i.e., dissolved organic carbon (DOC). In summary, our results suggest that OAE
efficiency as influenced by $CaCO_3$ precipitation not only depends on seawater composition but also on the physical properties
of the alkaline feedstock used.



**1. Introduction**

The concentration of carbon dioxide ($CO_2$) in the atmosphere has been in a relatively narrow band from ~180 to ~280
ppmv for the last 800,000 years, but has risen rapidly over the last 250 years to approximately 420 ppmv today (Lüthi et al.,
2008, Monnin et al., 2001, Siegenthaler et al., 2005). This is the result of increasing utilisation of fossil fuels, cement production
and land-use change, driving subsequent global climate change (IPCC, 2021). While about 42% of $CO_2$ emissions remain in
the atmosphere, and are mainly responsible for global warming, about 26% are currently absorbed by the oceans, leading to
ocean acidification (Friedlingstein et al., 2022, IPCC, 2021). To mitigate the effects of ocean acidification and slow down the
increase in Earth's global temperature, $CO_2$ reduction efforts are not sufficient and the use of carbon dioxide removal (CDR)
strategies have become necessary as a supplement to emission reduction (Hoegh-Guldberg et al., 2019).
One emerging marine CDR approach is ocean alkalinity enhancement (OAE). Over long timescales, the natural $CO_2$-
facilitated weathering of alkaline rocks supplies alkalinity to the oceans, influencing its $CO_2$ uptake potential and storage. OAE
builds upon this weathering feedback in the Earth System and can be accomplished by actively spreading pulverized alkaline
minerals in and around marine environments or electrochemically removing acidity from seawater (Eisaman et al., 2023). In
both cases, the seawater total alkalinity (TA) is increased thereby increasing the storage capacity of seawater for atmospheric
$CO_2$ (GESAMP, 2019, Kheshgi, 1995). On local scales around where the OAE perturbation is made, the increase in alkalinity
and pH may also mitigate ocean acidification (Hartmann et al., 2013).
Recent studies have investigated the carbonate chemistry changes following OAE, and a major outcome was the risk
for runaway calcium carbonate ($CaCO_3$) precipitation (Fuhr et al., 2022, Hartmann et al., 2023, Moras et al., 2022). There are
several inorganic $CaCO_3$ precipitation mechanisms that have been described in the literature (Morse et al., 2007, Pytkowicz,
1965). $CaCO_3$ can precipitate homogeneously in the absence of solid or soluble organic and inorganic particles, pseudo-
homogeneously in the presence of organic surfaces, and heterogeneously in the presence of mineral solids (Marion et al.,
2009). The key parameter that governs whether precipitation occurs is the calcium carbonate saturation state ($\Omega$), which is
calculated from seawater $Ca^{2+}$ and $CO_3^{2-}$ concentrations as:

$$\Omega = \frac{[Ca^{2+}]\,[CO_3^{2-}]}{K_{sp}}$$


where $[Ca^{2+}]$ and $[CO_3^{2-}]$ are the concentrations of calcium and carbonate in solution, respectively, and $K_{sp}$ the solubility
product of $CaCO_3$ in the solution. $\Omega$ is therefore closely related to the composition of the solution and its salinity, but is also
highly temperature dependent (Zeebe and Wolf-Gladrow, 2001). For aragonite, the $CaCO_3$ morphotype that inorganically



precipitates in modern seawater, the saturation state ($\Omega_A$) has to be higher than 12.3 for pseudo-homogeneous precipitation to
occur in water with a salinity of 35 and at 25 ºC (Marion et al., 2009). Homogeneous precipitation will occur at much higher
$\Omega_A$ values, while heterogeneous precipitation will occur already at much lower $\Omega_A$ but depends on the actual lattice
compatibility of $CaCO_3$ for the mineral particles present (Morse et al., 2007, Zhong and Mucci, 1989). Another important
aspect is that once precipitation becomes measurable, it will continue in a "runaway" fashion, i.e., quickly ramping up until it
slows down once $\Omega_A$ gets closer to 1 again.
Several studies have reported such behaviour upon mineral alkalinity addition (Fuhr et al., 2022, Hartmann et al.,
2023, Moras et al., 2022) with critical threshold of $\Omega_A$ of ~7.0 for the two calcium based OAE minerals of calcium oxide –
CaO – and calcium hydroxide – $Ca(OH)_2$ – and report precipitation stopping at $\Omega_A$ values of 1.8-2.0 (Moras et al., 2022).
Precipitation has also been observed for magnesium-based minerals such as brucite or reagent grade magnesium hydroxide –
$Mg(OH)_2$, but actual thresholds have not been determined (Hartmann et al., 2023). Furthermore, the effect of grain size,
determining factor of the surface area available for mineral dissolution and $CaCO_3$ precipitation, has not been studied.
Similarly, the effect of potential $CaCO_3$ precipitation inhibitors such as seawater magnesium (Mg) concentrations, governed
by salinity, and dissolved organic carbon (DOC), are relatively unknown (Chave and Suess, 1970, Millero et al., 2001, Pan et
al., 2021, Zhong and Mucci, 1989). This study focuses on the impact of $Mg(OH)_2$ grain size on its dissolution kinetics in
natural seawater, as well as the impact of salinity. Furthermore, the subsequent runaway $CaCO_3$ precipitation that is triggered,
and its kinetics are reported. Finally, the effect of increased [Mg] and [DOC] in seawater on the $CaCO_3$ precipitation process
is explored.

**2.    Material and methods**
**2.1. Seawater collection and experimental setup**
Seawater was collected in Broken Head, New South Wales, Australia (25°42'12'' S, 153°37'03'' E) using 25 L jerry
cans, about 200 m from the shore to avoid sampling sand and suspended particles. The collected seawater was stored in the
dark at 4 °C for three days to reduce microbial activity and allow particles to settle to the bottom, facilitating filtration. The
entire contents of the jerry cans were then sterile filtered using a peristaltic pump and a 0.2 μm Whatman Polycap 75 AS filter,
before being stored in cleaned and autoclaved 25 L polycarbonate bottles. Prior to conducting the experiments, each seawater
batch was equilibrated to laboratory air $pCO_2$ by bubbling them with $H_2O$-saturated air for at least a week (Moras et al., 2023).
This ensured comparable starting conditions for the various experiments. All experiments utilised reagent grade $Mg(OH)_2$
(>98%, kindly supplied by Atlas Materials) which had been ground in a Pulverizer laboratory mill.




### 2.2. Grain size and salinity experiments

Approximately 1.5 litres of seawater were placed in a clean 2 L borosilicate 3.3 beaker, surrounded by a water jacket
set to 21 °C and controlled by a tank chiller line TK-1000. A floating lid with three ports was placed on the water surface,
allowing for concurrent $Mg(OH)_2$ addition, pH measurement and water sampling. Upon $Mg(OH)_2$ addition, the seawater was
incubated for 18 hours to allow for full $Mg(OH)_2$ dissolution. Thereafter the beaker content was transferred to a clean 1 L
borosilicate 3.3 Schott bottle which was tightly closed without any headspace to minimise $CO_2$ ingassing. The bottle was
placed on a stirring platform at 200 rpm in the dark, at room temperature (24.8 ±1.3 °C). All grain size and salinity treatments
were run in triplicates for up to 34 days.
For the grain size experiments, three grain size ranges were produced using two stainless steel sieves with 63 µm and
180 µm mesh sizes. The medium range, i.e., 63-180 µm, was also used for the salinity experiments at ~36, ~28 and ~20. The
lower salinity seawater was produced by mixing natural seawater with MilliQ water. Exact salinities were determined on 200
mL of seawater sample equilibrated to room temperature in a gas tight polycarbonate container, by measuring conductivity
and temperature with a 914 pH/conductometer, and converted to salinity using the 1978 practical salinity scale (Lewis and
Perkin, 1981). For all experiments, $Mg(OH)_2$ additions were adjusted to yield an $\Omega_A$ of ~9 (Table 1) to allow for a significant
TA increase and secondary $CaCO_3$ precipitation, based on previously found thresholds for CaO and $Ca(OH)_2$ (Moras et al.,

2022).

In all the experiments, the first 18 hours of reaction were monitored by measuring the pH on the free scale ($pH_F$) with
an Aquatrode Plus with Pt1000 (Metrohm) connected to an 888 Titrando (Metrohm), before transferring the content of the 2
L beaker into the clean 1 L Schott bottles. A sample for TA and DIC measurements was taken before $Mg(OH)_2$ addition, and
after the 18 hours. The temperature and $pH_F$ were then recorded twice a day until a sudden drop in $pH_F$ was observed, linked
to $CaCO_3$ precipitation. A new sample for TA and DIC measurements was then taken. The time at which $CaCO_3$ runaway
precipitation was deemed to have started was considered to be the last stable $pH_F$ measurement before the sudden drop. TA
and DIC samples were taken at varying intervals during $CaCO_3$ precipitation (see figures) to cover most of the $CaCO_3$
precipitation process, and at least 300 mL of water was reserved for two TA and DIC samples at the end of the experiment.
Between 9 and 10 TA and DIC samples per experiment were collected to monitor the changes in DIC and TA overtime. Their
decrease in a 2:1 ratio was further used to reconstruct TA and DIC from pH measurements in the experiments on the effect of
Mg and DOC on $CaCO_3$ precipitation (see below for details).





**Table 1: Summary of the main experimental parameters for each of the incubations investigating the salinity and grain size effects on Mg(OH)₂ dissolution and CaCO₃ precipitation kinetics.**

| Experimental details | TA increase (µmol kg⁻¹) | Maximum Ω_A reached | Days of stable TA | Overall TA loss (µmol kg⁻¹) | Overall DIC loss (µmol kg⁻¹) | Final Ω_A |
|---|---|---|---|---|---|---|
| *Salinity effect on Mg(OH)₂ dissolution and CaCO₃ precipitation kinetics* | | | | | | |
| *Salinity 36* | | | | | | |
| Rep 1; Rep 2; Rep 3 | 555.5; 500.4; 534.9 | 9.23; 8.96; 9.16 | 10; 12; 9 | 1009.8; 1013.9; 1068.5 | 414.8; 477.4; 467.7 | 2.04; 1.95; 1.84 |
| Mean ± St. Dev. | 530.3 ±27.8 | 9.12 ±0.14 | 10.33 ±1.53 | 1030.8 ±32.8 | 456.7 ±27.9 | 1.94 ±0.10 |
| *Salinity 28.47* | | | | | | |
| Rep 1; Rep 2; Rep 3 | 618.7; 660.9; 615.8 | 9.18; 9.48; 9.29 | 6; 6; 4 | 1060.9; 1104.8; 1096.8 | 487.0; 494; 529.5 | 1.74; 1.68; 1.63 |
| Mean ± St. Dev. | 631.8 ±25.3 | 9.32 ±0.16 | 5.33 ±1.15 | 1087.5 ±23.4 | 503.5 ±22.8 | 1.68 ±0.05 |
| *Salinity 20.38* | | | | | | |
| Rep 1; Rep 2; Rep 3 | 575.9; 591.2; 605.3 | 8.26; 8.49; 9.14 | 2; 2; 1 | 899.3; 963.3; 1062.9 | 481.4; 522.8; 603.6 | 1.54; 1.51; 1.50 |
| Mean ± St. Dev. | 590.8 ±14.7 | 8.63 ±0.45 | 1.67 ±0.58 | 975.2 ±82.4 | 535.9 ±62.1 | 1.52 ±0.02 |
| *Grain size effect on Mg(OH)₂ dissolution and CaCO₃ precipitation kinetics* | | | | | | |
| *Small Grain size < 63 µm* | | | | | | |
| Rep 1; Rep 2; Rep 3 | 422.9; 447.5; 412.1 | 8.60; 8.48; 8.22 | 7; 4; 3 | 1019.3; 1021.9; 988.3 | 562.2; 547.3; 550.6 | 2.06; 2.16; 2.14 |
| Mean ± St. Dev. | 427.5 ±18.1 | 8.43 ±0.20 | 4.67 ±2.08 | 1009.8 ±18.7 | 553.4 ±7.8 | 2.12 ±0.05 |
| *Medium Grain size 63 – 180 µm* | | | | | | |
| Rep 1; Rep 2; Rep 3 | 555.5; 500.4; 534.9 | 9.23; 8.96; 9.16 | 10; 12; 9 | 1009.8; 1013.9; 1068.5 | 414.8; 477.4; 467.7 | 2.04; 1.95; 1.84 |
| Mean ± St. Dev. | 530.3 ±27.8 | 9.12 ±0.14 | 10.33 ±1.53 | 1030.8 ±32.8 | 456.7 ±27.9 | 1.94 ±0.10 |
| *Large Grain size > 180 µm* | | | | | | |
| Rep 1; Rep 2; Rep 3 | 368.9; 272.3; 412.6 | 8.41; 7.92; 8.72 | 3; 3; 2 | 1032.8; 980.7; 1103.1 | 606.1; 661.4; 647.5 | 1.89; 1.90; 2.02 |
| Mean ± St. Dev. | 351.3 ±71.8 | 8.35 ±0.40 | 2.67 ±0.58 | 1038.9 ±61.4 | 638.3 ±28.8 | 1.93 ±0.07 |




**2.3. Manipulation of dissolved inorganic carbon and magnesium**

The seawater dilution by MilliQ to decrease salinity also decreased the concentration of various seawater components, such as Mg and DOC concentrations. To disentangle a potentially general effect of salinity on $Mg(OH)_2$ dissolution and secondary precipitation kinetics from reductions in Mg and DOC concentrations, two additional experiments were designed. In the first, the experiments at a salinity of 20 were replicated, but the Mg concentration was increased to a concentration representative for a salinity of 35, i.e., 52.8 mmol kg$^{-1}$ (Dickson et al., 2007), by magnesium chloride ($MgCl_2$) addition from a 3 M stock solution (molarity verified by inductively coupled plasma mass spectrometer measurements). This experiment was run in triplicate. For the second experiment, a DOC-enriched seawater solution at the salinity of 20 was produced by ultrafiltration (molecular weight cut-off of 2,000 Daltons, Vivaflow200 Hydrosart, Sartorius). A DOC gradient was then created in five bottles by mixing the DOC-enriched salinity 20 seawater with the MilliQ-diluted seawater. The DOC concentrations ranged from approximately 120 µmol kg$^{-1}$ to approximately 325 µmol kg$^{-1}$.

In both the Mg and DOC experiments, dissolution and secondary $CaCO_3$ precipitation kinetics were mainly monitored by pH$_F$ measurements, although a sample for TA and DIC was also taken before $Mg(OH)_2$ addition and at the end of each treatment. These samples, coupled to the pH$_F$ measurements, allowed the changes in TA and DIC to be estimated over time. The reconstruction occurred in two steps, where the increase in pH following $Mg(OH)_2$ was assumed to be linked to an increase of TA at constant DIC. Then, any decrease in pH was assumed to be due to $CaCO_3$ precipitation, so the estimated TA and DIC loss after $Mg(OH)_2$ dissolution were decreasing in a 2:1 ratio, as observed in the salinity and grain size experiments. Finally, to account for $CO_2$ ingassing over time, the difference between estimated maximum TA and final measured TA was used as a proxy. Half of the difference, representing $CaCO_3$ precipitation, was used to estimate the theoretical DIC loss. Once compared to the final measured DIC, an ingassing rate was estimated.

**2.4. Analytical procedures**

The pH electrode was calibrated using three Metrohm buffer solutions (pH 4, 7 and 9), corresponding to a pH measurement on the free scale. TA analyses were conducted using a potentiometric titration with an 848 Titrino Plus, coupled to an 869 Compact Sample Changer from Metrohm. A 0.05M HCl solution with the ionic strength adjusted to 0.72 mol kg$^{-1}$ (representative for a salinity of 35) using NaCl was used as the titrant (Dickson et al., 2007). The DIC concentration was measured using an Automated Infra-Red Inorganic Carbon Analyzer (AIRICA) coupled to a LI-COR Li7000 Infra-Red detector (Gafar and Schulz, 2018). Both TA and DIC measurements were corrected against in-house reference material (previously calibrated against certified reference material), with measurement uncertainties of ±2.20 and ±1.98 µmol kg$^{-1}$ (Moras et al., 2023). $\Omega_A$ and carbonate chemistry speciation were calculated from TA and DIC, providing temperature and salinity measurements, using CO2SYS (Sharp et al., 2021).



For scanning electron microscopy (SEM), discrete samples of about 10 mL of TA enriched seawater were filtered

through 0.2 µm polycarbonate filters (Whatman Cyclopore). These filters were rinsed with 20 mL of MilliQ to remove salts

and dried overnight at 60 °C. Once dried, the filters were kept in a desiccator until analysis. The filters were attached to double-

sided carbon tabs and placed on aluminium mounts before being coated with gold. SEM analysis was performed using a

tabletop Scanning Electron Microscope TM4000 Plus from Hitachi, coupled to an Energy Dispersive X-Ray (EDX) Analyser,

allowing to determine the elemental composition of observed particles.

The concentration of the $MgCl_2$ stock solution was measured by inductively coupled plasma mass spectrometer (ICP-MS)

measurements using an Agilent 7700 ICP-MS, coupled to a laser ablation unit (NWR213, Electro Scientific Industries, Inc.).

Seawater reference materials from the National Research Council of Canada NASS-6 were used to correct the measurements.

The DOC concentration of the DOC-enriched stock solution was determined using a Thermo Fisher Flash Elemental Analyzer

after acidifying the sample with nitric acid (Carvalho, 2023).

**3. Results**

**3.1. Grain size effects on Mg(OH)₂ dissolution kinetics**

Three $Mg(OH)_2$ grain sizes were dissolved in seawater at a salinity of ~36 (Figure 1). The starting $pH_F$ was similar

for all incubations, with 8.11 ±0.03, 8.09 ±0.01 and 8.07 ±0.03, for the small (<63 µm), medium (63-180 µm) and large (>180

163       µm) grain sizes, respectively. Upon dissolution, $pH_F$ increased quite rapidly, reaching a maximum after about two hours for

the small particle size experiments, and about 6 to 8 hours in the medium and large particle size experiments (Figure 1). In

each incubation, a logarithmic trend in $pH_F$ was observed, with the dissolution being much quicker for smaller grain sizes.

After two hours, the maximum $pH_F$ recorded for the smaller grain size was 8.76 ±0.04, which continuously decreased to 8.68

±0.00 between 11 and 12 hours after $Mg(OH)_2$ addition. In contrast, the $pH_F$ for the medium and larger grain size increased to

8.72 ±0.00 and 8.68 ±0.03 after about eight hours and remained stable thereafter, respectively (Figure 1).



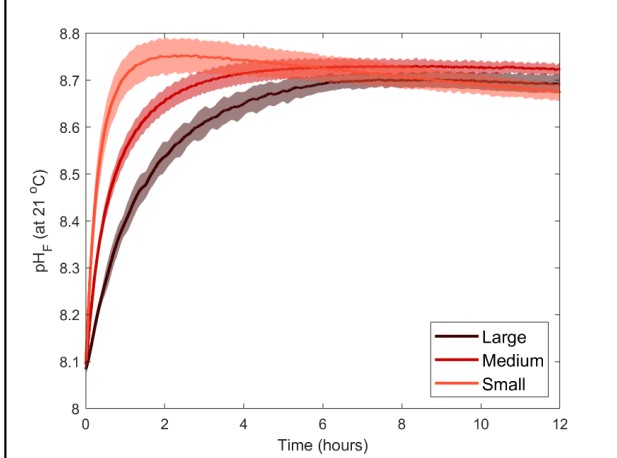

**Figure 1: Changes in $pH_F$ at 21 °C following dissolution of three $Mg(OH)_2$ grain sizes in natural seawater over 12 hours. Each grain size was run in triplicate, with the average presented as the solid lines and the standard deviation range as the transparent areas.**

### 3.2. Grain size effect on $CaCO_3$ precipitation kinetics

The pH increase was reflected by increasing TA, measured prior to the $Mg(OH)_2$ addition and 18 hours later, by about 430, 530 and 350 µmol kg$^{-1}$, in the small, medium and large grain size incubations, respectively (Figure 2). The TA remained stable for 3-7 days, 9-12 days, and 2-3 days before dropping in each grain size treatment (small, medium, large). In all incubations, TA concentrations decreased in a similar fashion, with a strong drop the first two days, before slowly decreasing for another week and stabilising. The overall TA loss for the duration of the experiments was ~1035 µmol kg$^{-1}$ in the medium and large grain size incubations, while the TA dropped by about 1010 µmol kg$^{-1}$ in the small grain size incubations (Table 1).

The changes in $\Omega_A$ followed a similar pattern as TA, increasing from ~2.8 on average to ~9.1 in the medium grain size incubation, and to ~8.4 in the small and large grain size experiments. $\Omega_A$ dropped at the same time as TA in the respective experiments, stabilising around ~2.0 in all experiments.

Finally, a small drop in DIC was observed after $Mg(OH)_2$ addition in all experiments, of about 80, 30 and 140 µmol kg$^{-1}$ in the small, medium and large grain size incubations, respectively. The DIC remained then relatively stable until the rapid TA drop, where the overall DIC drops for the small, medium and large grain size incubations were calculated at ~550, ~455 and ~640 µmol kg$^{-1}$, respectively. While TA and $\Omega_A$ remained stable after this drop, DIC slightly increased, particularly obvious in the medium and larger grain size incubations.



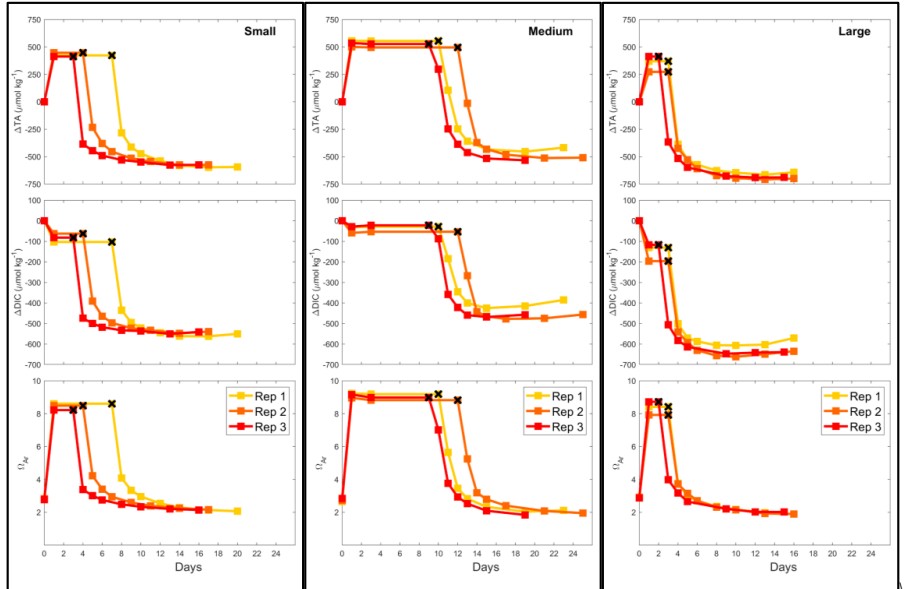

**Figure 2: Changes in TA, DIC and $\Omega_A$ during dissolution of three Mg(OH)$_2$ grain sizes in natural seawater over up to 25 days. Three replicates were conducted for each grain size and are represented in red, orange and yellow. The last stable TA and DIC conditions estimated by pH$_F$ measurements are represented by a black cross.**

**3.3. Salinity effect on Mg(OH)$_2$ dissolution kinetics**

To test the salinity effect on Mg(OH)$_2$ dissolution and CaCO$_3$ precipitation kinetics, three sets of experiments were conducted in three different salinities, i.e., 20.38, 28.47 and 35.80, and using medium grain size Mg(OH)$_2$. From here on the salinities 20.38, 28.47 and 35.80 will be referred to as salinities 20, 28 and 36, respectively. Similarly to the grain size experiments, the dissolution of Mg(OH)$_2$ occurred rapidly in all three salinities, with the maximum pH$_F$ being recorded after approximately 8 hours (Figure 3). Starting pH$_F$ were slightly different, recorded at 7.99 ±0.05, 8.06 ±0.01 and 8.09 ±0.01 in the salinity 20, 28 and 36 incubations, and increased to a maximum of 9.19, ±0.00, 8.91 ±0.00 and 8.72 ±0.00, respectively. In all incubations, similar logarithmic trends were observed for pH$_F$ (Figure 3).



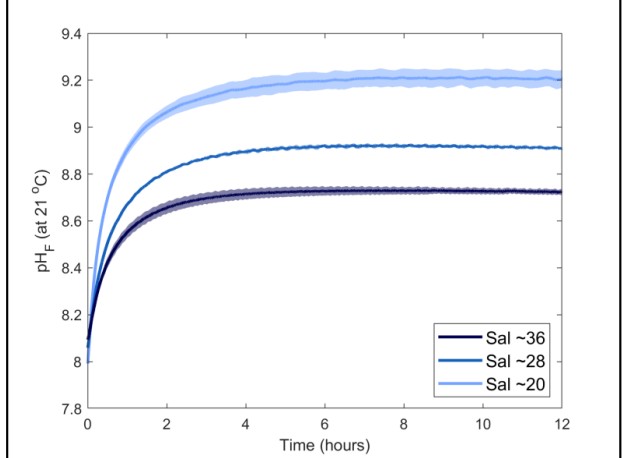

**Figure 3: Changes in $pH_F$ at 21 °C following $Mg(OH)_2$ dissolution in three different seawater salinities over 12 hours. Each salinity has been run in triplicate, with the average presented as the solid lines and the standard deviation range as the transparent areas. Please note that different maximum pH levels were reached because of increasing $Mg(OH)_2$ additions with decreasing salinity to reach a similar $\Omega_A$.**

### 3.4. Salinity effect on $CaCO_3$ precipitation kinetics

In all incubations, TA was increased as suggested by the $pH_F$ trends, by ~590, ~630 and ~530 µmol kg$^{-1}$ in the salinity 20, 28 and 36 incubations, respectively (Figure 4). The TA remained stable for different periods of time in each treatment; 1-2 days in the salinity 20 incubations, 4-6 days in the salinity 28 incubations, and 9-12 days in the salinity 36 incubations. Thereafter, TA dropped quickly the first two days in all incubations and stabilised quickly in the salinity 20 experiments. In the salinity 28 incubations, the TA slowly decreased over five days after the first strong drop and stabilised, while in the salinity 36 experiments, the TA decreased slowly after the initial drop over seven days before stabilising. The overall TA losses for salinities 20, 28 and 36 experiments were estimated at ~975, ~1090 and ~1030 µmol kg$^{-1}$, respectively (Table 1).

$\Omega_A$ values followed a similar pattern as TA in all experiments. The starting $\Omega_A$ were different, varying between 1.0 for the salinity 20 incubations to 2.0 and 2.8 for the salinity 28 and 36 incubations, respectively. Similarly, following $Mg(OH)_2$ additions, $\Omega_A$ quickly increased to reach 8.6, 9.3 and 9.1 with increasing salinity. Together with TA, $\Omega_A$ eventually started dropping, and then stabilised at different values, around 1.5 for a salinity of 20, around 1.7 for a salinity of 28 and around 2.0 for a salinity of 36.

Finally, DIC also decreased upon $Mg(OH)_2$ additions. An initial DIC drop was observed directly after $Mg(OH)_2$ additions of about 60 µmol kg$^{-1}$ at the lowest salinity and 30 µmol kg$^{-1}$ at the highest salinity. At a salinity of 28, a much smaller DIC drop was observed in one replicate. After a period of stable DIC conditions, DIC also dropped in a similar fashion



as TA, with an overall DIC loss of about 535, 505 and 455 µmol kg$^{-1}$ from the lower to higher salinity incubations. While no
DIC increase was observed towards the end of the experiment in the salinity 36 incubations, strong DIC increases were
observed in the salinity 28 incubations and even more prominent ones in the salinity 20 incubations.

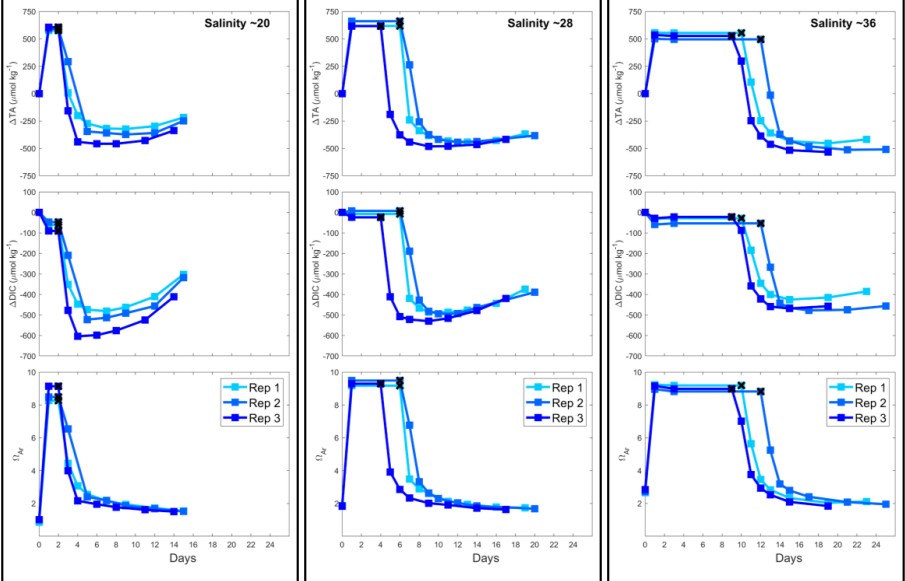


**Figure 4: Changes in TA, DIC and $\Omega_A$ during Mg(OH)$_2$ dissolution in three different salinities over up to 25 days. Three replicates**
**were conducted for each salinity and are represented in shades of blue. The last stable TA and DIC conditions estimated by pH$_F$**
**measurements are represented by a black cross.**

**3.5. Magnesium and DOC effect on CaCO$_3$ precipitation**
A similar pattern was observed for the salinity 20 experiments at natural and increased Mg concentrations, i.e., a rapid
increase in TA reaching a maximum on day one, followed by a steady decline over the next two weeks (Figure 5). The
maximum $\Delta$TA reached was slightly different, with about 600 µmol kg$^{-1}$ of TA increase in the salinity 20, and nearly 800
µmol kg$^{-1}$ in the salinity 20 + MgCl$_2$ incubations. Another interesting difference is the slower TA decrease with MgCl$_2$
compared to the salinity 20. After about 18 days, the lowest $\Delta$TA was reached while it only took about 6 days for the salinity
20 $\Delta$TA to reach the minimum. Similarly, DIC appeared to decrease less rapidly when MgCl$_2$ was present and $\Omega_A$ followed a
similar trend after the initial strong increase.
Out of the five DOC experiments, four incubations showed a drop in TA (Figure 5). Similar maximum $\Delta$TA were
reached in most experiments, with a $\Delta$TA of ~800 µmol kg$^{-1}$. However, in the incubation with ~120 µmol kg$^{-1}$, the TA increased



only by ~600 µmol kg⁻¹. Following this increase, TA decreased within a day in both 120 and 145 µmol kg⁻¹ DOC incubations,
and stayed stable until day 3 in incubations with 170 and 220 µmol kg⁻¹. These four incubations also show a similar levelling
pattern over time, even though it appears that in the higher DOC incubations, the total loss in TA was lower than for the lower
DOC incubations. ΔDIC also follow a similar trend to ΔTA, with an early drop at 120 µmol kg⁻¹ of DOC, a drop after one day
at 145 µmol kg⁻¹ of DOC, and a slow decrease from day 1 and a stronger drop on day 2 at 170 and 220 µmol kg⁻¹ of DOC. $\Omega_A$
followed a very similar pattern to ΔTA, with final $\Omega_A$ being higher in the experiments with higher DOC concentrations. Finally,
in the experiment with the highest DOC concentration, i.e., 325 µmol kg⁻¹, no drop in TA, DIC or $\Omega_A$ was observed (the
experiment was run for 42 days).

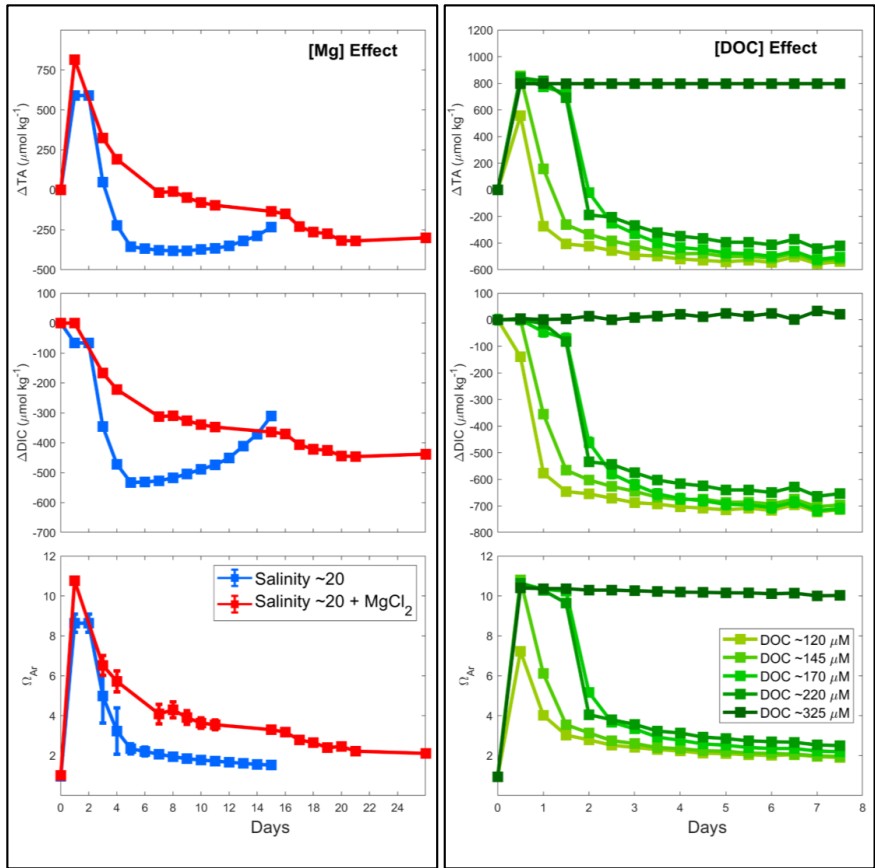


**Figure 5: Comparison of the reconstructed TA, DIC and $\Omega_A$ changes at 21 °C following Mg(OH)₂ addition in seawater with salinity**

**of 20 (blue), and in seawater with salinity 20 and Mg concentration equal to a salinity 35 (red), and in seawater with varying DOC**

**concentrations (green).**




**4. Discussion**
**4.1. Grain size and salinity effects on Mg(OH)$_2$ dissolution**
Maximum Mg(OH)$_2$ dissolution directly after its addition was negatively correlated with grain size (Figure 1, Figure
3). The smaller the grain size, the faster the maximum pH$_F$ is reached, indicative of complete dissolution. This can be explained
by the fact that smaller particles have a larger surface area per gram of material than larger ones. The increasing dissolution
rate with decreasing particle size is particularly noticeable in when TA changes were estimated by using the pH$_F$ data and
starting DIC measurements (Figure 6). Assuming a constant DIC over the first 30 minutes of reaction, i.e., no significant
CaCO$_3$ precipitation and/or CO$_2$ ingassing, TA can be reconstructed using CO2SYS. The maximum ΔTA reached with the
larger particle size occurred within 8 hours while it only took about 2 hours for the ΔTA to reach a maximum with small
particle size. The initial dissolution rate, i.e., within the first 30 minutes, was also significantly different between the various
grain sizes. The TA generation of smaller grain size particles was estimated at about 796.5 ±7.1 µmol of TA mg$^{-1}$ min$^{-1}$. The
medium particles dissolved about twice as slow over the first 30 minutes, estimated at 391.6 ±2.6 µmol of TA mg$^{-1}$ min$^{-1}$,
while the larger grain sizes dissolved more than four times slower, with about 168.7 ±6.9 µmol of TA mg$^{-1}$ min$^{-1}$. Another
important difference between the smaller grain size experiments and the two others is the constant decrease in pH$_F$ observed
right after reaching the maximum pH$_F$ value. This decrease in pH$_F$ can only be linked to either CaCO$_3$ precipitation, decreasing
TA and ultimately pH$_F$, or CO$_2$ ingassing, increasing the dissolved CO$_2$ concentration and ultimately decreasing the pH$_F$. The
constant and linear trend suggest that the latter is responsible for the decrease. If CaCO$_3$ was responsible for these pH$_F$ changes,
the changes would follow a similar pattern to a negative exponential function. This is due to the fact that the more CaCO$_3$
nucleate, the more surface becomes available for further nucleation (Zhong and Mucci, 1989). However, in our case, the
changes appear linear. Such a pattern is indicative of CO$_2$ ingassing at an early stage, i.e., before the ingassing starts plateauing,
dictated by the difference between atmospheric and seawater pCO$_2$. Such ingassing is also occurring in the other experiments,
but is likely hidden by the stronger pH$_F$ increase occurring during Mg(OH)$_2$ dissolution.
For salinity, we did not observe major differences in initial dissolution rates within the range of salinities tested, with
dissolution rates for salinities 36, 28 and 20 estimated at 391.6 ±2.6, 359.8 ±0.2 and 301.9 ±0.3 µmol of TA mg$^{-1}$ min$^{-1}$,
respectively. Overall, TA generation potential of smaller grain size Mg(OH)$_2$ (<63 µm) at a salinity 36 was similar to that of
Ca(OH)$_2$ (Moras et al., 2022) which was also sieved through 63 µm. Assuming the same molar TA generation potential, the
same maximum Ω$_A$ should have been reached. However, for Ca(OH)$_2$ it was ~7.4, while our small grain size Mg(OH)$_2$
incubations reached a maximum Ω$_A$ of ~8.4. Such a difference is likely due to the difference in the starting conditions. In the
experiments shown here, the starting Ω$_A$ was ~2.8 while it was about ~2.5 in Moras et al. (2022). This is explained by the
difference in the starting water composition and salinity, ultimately affecting the final Ω$_A$ despite similar TA increases.
However, dissolution kinetics appear to differ between the minerals, with Ca(OH)$_2$ dissolving within 20-30 minutes while it
took two hours for Mg(OH)$_2$. These two minerals still dissolve at a relatively quick pace compared to other OAE feedstocks,



for instance olivine (Montserrat et al. 2017). Olivine took much longer to dissolve, with a maximum increase in pH recorded
of ~0.15 units within 4-9 days. $Ca(OH)_2$ and $Mg(OH)_2$ additions required ~20 mg of materials, while to obtain such olivine
results, more than 30 g of olivine were added per kg of filtered seawater, meaning that the TA generation potential is several
orders of magnitude lower.

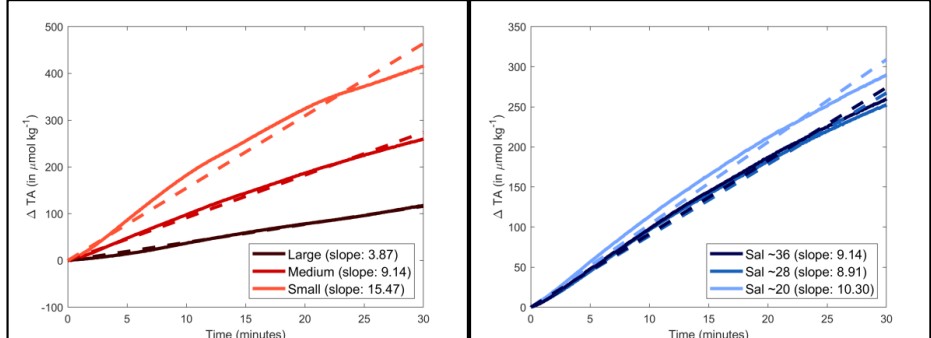

**Figure 6: Normalised changes in TA over the first 30 minutes following $Mg(OH)_2$ additions of three different grain sizes in natural**
**seawater (left) and in three different salinities (right). A linear fit was calculated and is represented by the dashed line, and each**
**slope is reported in the legend in between parentheses.**

### 4.2. Grain size and salinity effect on $CaCO_3$ precipitation

In all experiments, $Mg(OH)_2$ additions had been chosen to reach an $\Omega_A$ at which secondary $CaCO_3$ precipitation would
be expected based on our experience with CaO and $Ca(OH)_2$ (Moras et al., 2022). Based on our suspicion that $CaCO_3$ might
precipitate on magnesium-rich particles less easily than onto calcium-rich particles we chose a saturation state of ~9, slightly
higher than the level of ~7 observed for CaO and $Ca(OH)_2$ (Moras et al., 2022). Precipitation kinetics were similar for all grain
sizes, i.e., after the first precipitation was observed a new steady state was achieved in about two weeks. Precipitation
seemingly stopped at $\Omega_A$ values close to 2.0 in experiments with seawater at a salinity of 36, similar to observations made by
Moras et al. (2022) using CaO and $Ca(OH)_2$. For the smallest grain size, TA was stable for 3-7 days, which is longer than what
has been observed for CaO and $Ca(OH)_2$ at the same size (Moras et al., 2022). This could be related to higher lattice
compatibility of $CaCO_3$ for calcium-based minerals when it comes to precipitation onto mineral surfaces (Lioliou et al., 2007).
Interestingly, however, the rate at which $CaCO_3$ precipitated was similar for CaO and $Mg(OH)_2$, while $Ca(OH)_2$ took almost
twice as long to reach a new steady state (compare Figure 1 with Figure 2 in Moras et al., 2022).
TA remained stable for longer, i.e., 9-12 days with medium grain size. However, similarly to the smaller grain size
experiments, the TA was also less stable with the larger grain size, i.e., 2-3 days. As such, there appears to be an optimum



grain size for keeping TA stable for longer. To explain this, there must be two opposing processes at work. As discussed
earlier, smaller particles have larger surface area per gram of material than larger ones, i.e., smaller particles in our experiments
had on average more than 23 times the area of larger particles for the same amount of material, assuming round particles of 63
and 180 μm, respectively. Hence, heterogeneous precipitation will be quicker for smaller particles (Zhong and Mucci, 1989).
In contrast, what could favour quicker precipitation for larger particles with smaller surface area remains to be understood.
Here, it could be higher pH levels and hence $\Omega_A$ that are reached at a particle's surface as of having a larger diffusive boundary
layer. Hence, pH and $\Omega_A$ levels are likely to be much higher and remain for longer due to the slower dissolution of larger
particles at the site of $CaCO_3$ nucleation, which positively affects $CaCO_3$ precipitation rates.
To our surprise, EDX analysis did not reveal significant magnesium concentrations in early precipitated aragonite
crystals, i.e., ~18 hours after $Mg(OH)_2$ addition. The presence of Mg could have been expected if $CaCO_3$ precipitated
heterogeneously onto $Mg(OH)_2$ particles (Figure 7). The absence of Mg after EDX analysis suggests that while some $Mg(OH)_2$
could have been used as a precipitation nuclei for $CaCO_3$ early on, it completely dissolved within the first 18 hours. Only the
freshly precipitation $CaCO_3$ would then remain in suspension, eventually acting as precipitation nuclei for runaway $CaCO_3$
precipitation. Finally, it is interesting to highlight that some traces of early aragonite crystals were present in all experiments,
and that the needle-shaped crystals were two to three times smaller in the larger grain size experiments than those sampled at
the end of the medium grain size experiments (Figure 7). One explanation that supports the previously mentioned boundary
layer theory is that the larger grain size particles, dissolving at a slower pace, maintained a Mg-rich environment while $CaCO_3$
started nucleating. The presence of this Mg during nucleation could have ultimately prevented $CaCO_3$ to fully form as bigger
needle-like crystals. However, these are speculations that are hard to prove or disprove.

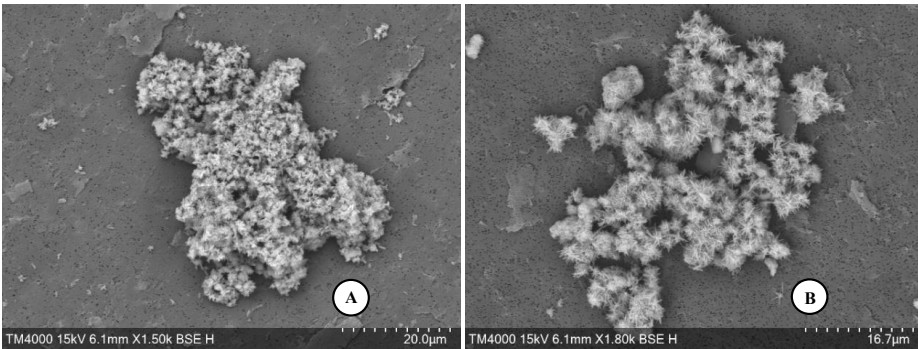

**Figure 7: SEM images of aragonite crystals, sampled ~18 hours after larger Mg(OH)₂ grain size addition (A) and sampled at the end**
**of the medium grain size incubations (B).**



### 4.3. The role of dilution and potential effects of Mg and DOC concentrations

The role of Mg in inhibiting $CaCO_3$ nucleation is well known (Morse et al., 2007, Pan et al., 2021, Pytkowicz, 1965). Another known $CaCO_3$ nucleation inhibitor is organic matter, particularly dissolved organic matter (Chave and Suess, 1970). While the role of organic matter is not as well understood as Mg, both have been linked to a decrease in $CaCO_3$ nucleation and precipitation rates.

In our experiments involving dilution with MilliQ water, all dissolved components of the seawater were diluted, including Mg and DOC. Such decreases could explain the quicker $CaCO_3$ precipitation in the salinity 20 experiments compared to salinity 36, as lower Mg and DOC concentrations were not inhibiting precipitation as in the higher salinity treatments. To test this, a new salinity 20 batch was prepared in triplicate and Mg was added to raise the total Mg concentration to ~52 mmol $kg^{-1}$, similar to the Mg concentration in natural seawater at a salinity 35. The Mg increase did affect $CaCO_3$ precipitation kinetics as shown by changes in TA (Figure 5), being slightly slower and apparently reaching a new steady state at higher $\Delta TA$ and $\Omega_A$. Furthermore, it is important to highlight that despite $CaCO_3$ precipitation being triggered at a similar time, i.e., within 1 to 2 days, a difference was observed regarding the maximum $\Delta TA$ reached. In the salinity 20 + $MgCl_2$ experiments, the maximum $\Delta TA$ value was higher than the one in the salinity 20 experiments. This suggests that with a higher dissolved Mg concentration, less $CaCO_3$ is precipitated early on. Following this early precipitation, an overall slower precipitation rate is observed until reaching a steady state (Figure 5).

However, the slightly reduced $CaCO_3$ precipitation rate due to decreased Mg concentrations alone cannot explain such stark differences in TA stability between the salinity 36 and 20 experiments (Figure 4). It is most likely linked to both the decrease in Mg and DOC concentrations when diluting with MilliQ. The gradient of five salinity 20 replicates with increasing DOC concentrations clearly showed that secondary $CaCO_3$ precipitation could be delayed by modifying the DOC concentrations alone. For instance, secondary precipitation became already measurable after 12 hours at DOC concentrations of 120 µmol $kg^{-1}$, i.e., salinity 35 diluted to 20, but almost no secondary precipitation at a DOC concentration of 325 µmol $kg^{-1}$, i.e., about one and a half times higher than in the salinity 35. $CaCO_3$ precipitation was delayed by about two days when doubling DOC concentration, and completely prevented at even higher levels (Figure 5) within the timeframe of the experiment (1 week). Together, these data suggest that seawater DOC and Mg act in synergy when it comes to inhibiting $CaCO_3$ precipitation.

Another interesting finding was the new steady state reached after runaway $CaCO_3$ precipitation. In natural seawater at a salinity of 36, the equilibrated $\Omega_A$ was estimated around 2.0, which is about 0.8 units lower than the starting conditions (Figure 4). The decrease in $\Omega_A$ after runaway precipitation has important implications for OAE, as when $CaCO_3$ precipitates in a runaway fashion, seawater can become more acidic than it was prior to mineral dissolution and less able to sequester atmospheric $CO_2$ (Moras et al., 2022). While further work is required to understand these carbonate chemistry mechanisms at



lower salinities, we can note that after runaway precipitation in seawater at a salinity of 20, the final $\Omega_A$ was higher than the
starting one. Such a difference is likely due to the lower starting $Ca^{2+}$ concentration at lower salinity.

**5.    Conclusions**

One main objective of this research was to assess the dissolution of $Mg(OH)_2$ in seawater at varying salinity, and

using different mineral grain sizes, and report on the subsequent $CaCO_3$ precipitation kinetics. The dissolution of $Mg(OH)_2$ in
natural seawater occurred at a much faster rate when using grain sizes lower than 63 µm, due to the higher surface area in
contact with seawater. In contrast, bigger particles (>63 µm) took about four times as long to fully dissolve. In all experiments,
$CaCO_3$ precipitation occurred in a runaway fashion, i.e. after a period of seeming stability, TA decreased rapidly before a new
steady state was reached at which TA reached concentrations far lower than prior to the $Mg(OH)_2$ addition. A major finding
was that two processes occur during $CaCO_3$ precipitation in relation to grain size, one where the higher surface area of smaller
particles increases precipitation rates, while the second maintains a higher pH around larger particles due to a larger diffusive
boundary layer compared to smaller particles, which increases precipitation rates. Hence, there appears to be an optimum grain
size to minimise secondary $CaCO_3$ precipitation. The second objective of this research was to understand the role of salinity
on $Mg(OH)_2$ dissolution and $CaCO_3$ precipitation kinetics. While no obvious changes in dissolution were observed, $CaCO_3$
precipitation differed, with a quicker precipitation observed at lower salinities. The decrease in Mg concentrations was
identified as the root cause, although in our experiments it was also linked to a lowered DOC concentration, an artefact of low
salinity seawater preparation by dilution with MilliQ. Nevertheless, this highlights the importance of DOC in modifying $CaCO_3$
precipitation kinetics and hence, TA stability.

**Data availability**

All data will be made available upon acceptance of the manuscript by *Biogeosciences*.


**Author contributions**

CAM and KGS designed the initial experiments with inputs from TC and LTB. CAM ran all the experiments and

with the help of KGS designed the follow-up experiments with $MgCl_2$ and DOC. The ICP-MS analyses were performed by
CAM and RJB, while CAM and KGS performed the SEM analyses. The first draft of the manuscript was written by CAM
with inputs from KGS, and all co-authors have helped writing and reviewing the manuscript for submission.



**Competing interests**

At least one of the (co-)authors is a member of the editorial board of *Biogeosciences*.

**Acknowledgments**

We would like to sincerely thank Atlas Materials for providing the magnesium hydroxide. We are also thankful to Nick Ward for accommodating the use of the Scanning Electron Microscope, as well as Matheus Carvalho de Carvalho for the dissolved organic carbon analyses.

**Financial support**

This research is part of the PhD project of Charly A. Moras that is funded by a Cat. 5 – SCU Grad School scholarship from the Southern Cross University, Lismore, Australia. The ICP-MS analyses were made possible by Australian Research Council grants to Renaud Joannes-Boyau and Kai G. Schulz (grant no. LE200100022) and to Renaud Joannes-Boyau (grant no. LE120100201).



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
