# Peer review of "Effects of grain size and seawater salinity on magnesium hydroxide"

_EGUsphere, 2024_

## Author Comment (AC1)

**Overview and general comments:**

The manuscript of Moras et al. reports results from a laboratory study on magnesium hydroxide additions to seawater, examining the effects of mineral grain size, seawater salinity, and dissolved organic carbon concentrations on the dissolution of magnesium hydroxide and secondary calcium carbonate precipitation. Their results demonstrated that secondary CaCO3 precipitation was delayed the longest when using medium grain size Mg(OH)2, and they also found that secondary precipitation was accelerated at lower concentrations of Mg and DOC, both of which are known CaCO3 precipitation inhibitors. This is an important study with implications for optimizing the efficacy of ocean alkalinity enhancement.

The authors' conclusions regarding the effects of grain size, Mg, and DOC on delaying secondary precipitation are generally supported by the data, and the manuscript was straightforward to follow. Some major issues that need to be addressed include providing more details on the carbonate chemistry calculations (including uncertainty estimates—especially for Ω, which is an indicator for the secondary precipitation threshold), clearly distinguishing between calculated and measured CO2 parameters, and the unexplained DIC trends in some of the data (which the authors attribute to ingassing, but does not seem to be well-supported by the data). Additionally, the presentation of the some of the figures and tables can be improved. With minor revisions, I recommend publication.

**Major Comments:**

Carbonate Chemistry Calculations:

The authors need to provide full details on the carbonate chemistry calculations, such as the choice of constants used in the calculations (Lines 145-146). Additionally, uncertainties should be estimated and reported for calculated CO2 parameters using the uncertainty propagation routines available on CO2SYS. The uncertainty estimates are especially important when reporting values of Ω, which undoubtedly will inform future research on critical thresholds for CaCO3 precipitation. Without uncertainty estimates, the results in this study cannot be interpreted. In Table 1, the calculation uncertainty for Ω should be combined in quadrature with the standard deviation of the replicates.

Thank you for the comment. We indeed forgot to mention the constants used. This has been added in lines 154-156 (**of the TRACK CHANGES document**). Similarly, the uncertainties for calculated CO2SYS parameters have been added to the graphs and in table 1.

Calculated vs. Measured CO2 Parameters:

The text and figures are often confusing as to which CO2 parameters were measured and calculated. Furthermore, the authors do not always specify which input parameters were used in the calculations. In Lines 127-135, it appears that most of the DIC and TA values in the Mg and DOC experiments were calculated values (unlike in the salinity and grain size experiments where each experiment had 9 to 10 DIC and TA measurements). Fig. 5, however, does not make it clear which of the ΔDIC and ΔTA values were measured and calculated (and the input parameters used). Similarly, in Line 133, "the difference between estimated maximum TA and final measured TA" does not specify how TA was calculated.

Thank you for the comment. In the caption of Figure 5, the term "reconstructed" will be replaced to "calculated". Regarding the second part of the comment the information is given at line 136, where we specify that **"a sample for TA and DIC was also taken before Mg(OH)2 addition and at the end of each treatment"**, suggesting that these TA and DIC data have been measured the same way than other TA and DIC samples, and used for the reconstruction.

pH measurements on the "free" scale:

The authors report pH measurements on the "free" scale based on pH electrode measurements calibrated with Metrohm buffer solutions (pH 4, 7, and 9). As the calibration solutions are low-ionic strength solutions, there will be large liquid junction potential errors when making measurements in seawater solutions at high ionic strength, and it is difficult to estimate the uncertainty. While the electrode measurements may still be useful in reporting changes in pH, it seems problematic to use the data to calculate other CO2 parameters in seawater. One solution may be to calibrate the electrode measurements against pH calculated from measured DIC and TA whenever these measurements coincide. The authors should also make clear which parameters were calculated with pH and how the uncertainty in the pH measurements may affect the interpretation of their results.

Thank you for the comment. To perform accurate CO2SYS calculations, the scale on which the pH is measured needs to be known. Based on the buffer used, the pH probe was calibrated on the free scale. This was further confirmed by comparing the measured pH with estimated $pH_F$ from CO2SYS using measured TA and DIC. After reviewing all data, the measured pH was identical to the calculated $pH_F$ from CO2SYS. Therefore, this allowed us to report the pH values as $pH_F$. Regarding the difference between calculated and measured parameters, this information is given in paragraphs 2.3 and 2.4. All samples for TA and DIC were measured, expect when clearly stated as for the Mg and DOC experiments (lines 137 to 143).

Maximum ΩA attained (Line 288): The authors attribute differences in the maximum ΩA attained in their Mg(OH)2 experiments and previous studies with Ca(OH)2 to differences in the starting water composition. It is difficult for the reader to confirm this without information provided on the starting water composition. I suggest visualizing the results of this and previous studies with Ca(OH)2 on a DIC and TA diagram with ΩA isolines, showing the initial water composition and final composition at maximum ΩA.

Thank you for pointing this out, we agree that this can be hard for the reviewer. We have reviewed the text, and it now reads (lines 299-303): **"Such a difference is likely due to the difference in the starting conditions and experimental settings. In the experiments shown here, the starting ΩA was ~2.8 while it was about ~2.5 in Moras et al. (2022). This is explained by the difference in the starting water composition and salinity, ultimately affecting the final ΩA despite similar TA increases. Furthermore, higher amounts of Mg(OH)2 were added compared to Moras et al. (2022), leading to a higher ΩA and a higher theoretical ΔTA, if no early CaCO3 nucleation occurred."**

Secondary CaCO3 precipitation thresholds:

The authors chose ΩA ~ 9 based on their hypothesis that the secondary precipitation threshold would be higher when adding Mg(OH)2 compared to the threshold when adding CaO and Ca(OH)2. However, the threshold was not directly investigated in this study. It would be worth discussing how future research might better constrain the threshold for Mg(OH)2 additions and how secondary precipitation might be mitigated when adding Mg(OH)2 vs. CaO and Ca(OH)2. Is it more advantageous to optimize the grain size of the feedstock or to stay below the critical Ω threshold?

Thank for this comment. We agree that the threshold not being directly investigated, a discussion would be a good addition to the manuscript. We added (lines 404-408): **"Such pattern was also observed for Ca-rich minerals as well, but at lower ΩA. While further research is required to precisely determine the critical ΩA for both Ca- and Mg-rich minerals, the longer time for CaCO3 runaway precipitation to be initiated and the overall higher ΩA may suggest that Mg(OH)2 is a safer alkaline feedstock for OAE. Nevertheless, staying under critical ΩA should remain the primary target of any OAE"**.

Regarding the second part of the question, we believe that staying below the critical $\Omega_A$ threshold is and remain the more suitable option. However, most OAE research stresses out the importance of efficient grinding. This manuscript helps understanding that while dissolution rates can be effectively increased with smaller grain size, the higher surface area is a problem that need to be monitored as well.

CO2 ingassing in experiments:

In many of the experiments (see Fig. 2, 4, 5, Line 187, Line 227), the data show the DIC increasing after the rapid drop in DIC and TA from CaCO3 precipitation, yet the cause of this trend is not well-explained. In Fig. 1, the data show the pH of the small grain size experiment continuously decreasing after reaching its maximum, which the authors attribute to ingassing. Yet, this was not apparent in the medium and large grain size experiments. In Sect. 4.1, the authors claim that ingassing is likely occurring in the other experiments as well, but masked by the stronger pH increase during dissolution. However, their explanation seems to contradict Fig. 1, which shows the strongest pH increase during dissolution for the smallest grain size.

Thank you for the comment. We agree with the points the reviewer made and edited the text accordingly (lines 291-292 and 338-345). We explained that the $CO_2$ ingassing, while occurring in all experiments, is only clearly visible in the smaller grain size as in the other experiments, the slower dissolution led to a longer $pH_F$ increase overtime. We therefore believe that over the last 2-4 hours of $pH_F$ measurements, the changes appear stable while it is likely that the $pH_F$ goes down due to ingassing while increasing due to the last grains of $Mg(OH)_2$ dissolving overtime.

What was the pCO2 of the seawater samples relative to the laboratory atmospheric pCO2, and is this consistent with ingassing?

The starting $pCO_2$ was consistent in all experiments, with a starting $pCO_2$ of about 420.61 ±28.59 µatm across experiments. We added this information in lines 84-85. Yes, $pCO_2$ was consistently lower than atmospheric $CO_2$ at the end of the experiments, with:

- Salinity 35: 389.51 ±10.01 µatm
- Salinity 28: 258.51 ±8.30 µatm
- Salinity 20: 116.45 ±15.90 µatm
- Large grain size: 312.24 ±26.15 µatm
- Medium grain size: 389.51 ±10.01 µatm
- Small grain size: 307.27 ±4.87 µatm

One weakness in the experimental design was that there were no control samples (subjected to the same headspace conditions from the periodic sampling but not the Mg(OH)2 addition) from which the magnitude of gas exchange can be directly assessed. The authors should address how they can be certain that the changes in DIC not attributable to CaCO3 precipitation are necessarily due to gas exchange.

Thank you for the comment. We understand the reviewer worries, however, any decrease in DIC could only be linked to $CaCO_3$ precipitation as the seawater $pCO_2$ never exceeded the atmospheric one. Therefore, if DIC is removed from the water column, it cannot be from degassing but only through $CaCO_3$ precipitation. Similarly. Any increase in DIC in seawater can only come from $CO_2$, as carbonates do not dissolve in seawater. Furthermore, the lower seawater $pCO_2$ coupled to the increase in DIC can only come from $CO_2$ ingassing, rounding up the carbonate chemistry changes.

**Minor Comments:**

Line 21: Change "higher alkalinity/pH" to "higher alkalinity and pH."

This has been edited.

Line 83: What was the laboratory pCO2?

This was changed (lines 84-85) to **"This ensured comparable starting conditions for the various experiments, with a calculated starting pCO2 of 420.6 ±28.6 µatm in all experiments."**

Line 101: ΩA ~ 9 is not the threshold for CaCO3 precipitation when adding CaO and Ca(OH)2. It would be clearer to explain early on that ΩA ~ 9 was selected based on the suspicion that the precipitation threshold for CaCO3 precipitation when adding Mg(OH)2 may be higher than the ΩA ~ 7 threshold observed when adding CaO and Ca(OH)2. This was explained later in Lines 304-305.

This was added lines 101-104: **"For all experiments, Mg(OH)2 additions were adjusted to yield an ΩA of ~9 (Table 1) to allow for a significant TA increase and secondary CaCO3 precipitation, based on previously found thresholds for CaO and Ca(OH)2 , and with the assumption that the CaCO3 inhibition role of Mg2+ requires a higher ΩA for CaCO3 precipitation within days (Moras et al., 2022)"**

Line 109: Please reference the specific figures in "see figures."

This was changed (line 116) to **"(see figures 2 and 4)"**

Lines 110-111: What was the sample size for the TA and DIC measurements? The description in these two lines sounds like ~150 mL of sample was used for each TA and DIC measurement. However, if 10 samples were taken for TA and another 10 samples for DIC measurements (as suggested in Line 111), there would not be enough sample (1.5 L total). It sounds like most of the sample was consumed by the end of the experiment. Please clarify the measurement process and address how the increasing headspace over the course of the experiment may contribute to the instability of the DIC.

We understand the comment. However, the first two samples were taken from the reaction vessel with a floating lid, described in lines 89-90, where 1.5L of seawater was used. The whole content of the beaker was then transferred to 1L Schott bottles. However, while these bottles are labelled as

1L, they actually can contain slightly more, with about 1.2L when filled to the top. These 1.2L allowed us to collect up to 8 more samples for TA and DIC analyses.

Table 1: The organization of the table needs to be improved for clarity.

The Table 1 has been reworked and cleared from unused information. The table is divided in 4 blocks, with the experiment details, starting conditions, conditions at the peak of ΔTA and final conditions.

Firstly, the initial carbonate composition (e.g., DIC and TA) of the samples should be reported, not just the changes in DIC and TA.

Thank you for the comment, these values are now reported in table 1 for further clarity.

The first three columns after "Experimental details" pertain to the dissolution phase of the experiment and the next three columns pertain to the secondary precipitation phase. It would be helpful to label these columns as such.

Thank you for the comment, we have added three headings: "Starting Conditions", "Conditions after Full Dissolution", and "End Conditions".

Without reading the manuscript text, "Days of stable TA" is an unclear description. Consider including a short explanation in the table caption.

The following statement has been added to the caption (line 123): **""Days of stable TA" encompasses the time between maximum ΔTA recorded and the start of CaCO3 runaway precipitation."**

Line 176: The increase in TA for the small grain size experiment was ~428 umol/kg according to Table 1.

Thank you for pointing this out, however, we decided to report rounded number to the closest multiply of 10, i.e., 430 instead of 427.48, 530 instead of 530.27 and 350 instead of 351.25. for consistency, we preferred keeping the value as "430".

Fig. 2: Please specify which input parameter pairs were used to calculate DIC and TA from pH.

Thank you for the comment, we have added this information at lines 154-156.

**Panel figures (Fig. 2, Fig. 4, and Fig. 5):**

In general, the panel figures need to be improved for visibility as the font size on the axes may be too small.

The figures have been reprocessed and edited to increase the font size.

Why are the pH measurements not included?

$pH_F$ measurements were of prime interest at the start of the experiments (as per Figures 1 and 2). However, for $CaCO_3$ precipitation, TA and DIC measurements are more suitable since while $CaCO_3$ precipitate, $pH_F$ would decrease. At the same time, $CO_2$ ingassing can still occur, enhancing this $pH_F$ decrease. Therefore, the use of $pH_F$ for the second part of the experiment was not deemed relevant and TA and DIC were preferred overall.

Line 244: The text should be 120 µmol kg-1 DOC.

Thank you for pointing this out. The text has been edited accordingly (line 254).

Fig. 5. Please explain in the caption what the error bars represent in this and other figures.

Thank you for the comment. The following statement has been added to the caption (line 267-268): **"Values reported in the [Mg] Effect graphs represent the average of triplicate experiments run at salinity 20 and salinity 20 + MgCl2, with standard deviations represented by the error bars."**

Line 273: Reference Fig. 1.

Thank you for pointing this out. The text has been edited accordingly (line 284).

Line 275: The text should be "If CaCO3 precipitation."

Thank you for the comment, this has been edited (line 286).

Fig. 6: How were the TA data normalized?

Thank you for the comment. The data were normalised by reporting the changes in calculated TA (using the starting measured TA, DIC and salinity conditions, and the $pH_F$ measurements to reconstruct TA changes) from the start of the experiments over the first 30 minutes. This allowed to remove the effect of salinity and the different starting conditions (particularly visible in figure 3 due to low salinity) for better comparison.

Line 380-382: The mechanism of CaCO3 precipitation in relation to grain size was not demonstrated in this study, so this conclusion should be stated as a hypothesis rather than a finding.

Thank you for pointing this out. The text has been edited accordingly (line 404-407): **"A major finding was that two processes seem to occur during CaCO3 precipitation in relation to grain size, one where the higher surface area of smaller particles could increase precipitation rates, while the second could may maintain a higher pH around larger particles due to a larger diffusive boundary layer compared to smaller particles, which increased precipitation rates."**

---

## Author Comment (AC2)

Marine carbon dioxide removal, and ocean alkalinity enhancement in particular, have the potential to make a significant contribution to the world's efforts toward avoiding the worst effects of climate change. Mg(OH)2 is a promising alkalinity source, but understanding its dissolution kinetics and how to avoid runaway precipitation of CaCO3 that removes more alkalinity than was added is key to realizing this promise. I recommend "Effects of grain size and seawater salinity on magnesium hydroxide dissolution and secondary calcium carbonate precipitation kinetics: implications for ocean alkalinity enhancement" for publication, but only after the authors address my comments and questions listed below.

Abstract, lines 21-22: "while the slower dissolution of bigger grain size maintained a higher alkalinity/pH at the surface of particles, increasing CaCO3 precipitation rates and making it observable much quicker than for the intermediate grain size" Why would larger size have this effect? in the direction normal to the grain face, should be same in all cases, but is the argument lateral transport makes a parabolic alk/pH distribution with peak at center of grain face?

Thank you for the comment. We agree that the TA/pH would be consistent in the direction normal to the grain face. However, the larger grain sizes, taking longer to dissolve, provide high TA/pH environments for longer around the particles. This maintain of such high levels for longer than with small particles could allow for (pseudo-)homogeneous precipitation to occur.

Line 34: "To mitigate the effects of Ocean Acidification" No CDR process will meaningfully mitigate OA over a global scale, only local scales. I suggest changing to "To locally mitigate..."

This has been edited accordingly (line 34, **of the TRACK CHANGES document**).

Line 83: "laboratory air pCO2" What was the lab CO2 value? if H2O saturated, what was the temperature?

Thank you for the comment. The laboratory air was not recorded at the exact time, but the experiments run by Moras et al. (2023) occurred at a similar time and they recorded a laboratory $pCO_2$ of 479.1 ± 4.3 µatm. The laboratory temperature was regulated by air conditioning, set to 21 °C.

Line 95: "For the grain size experiments, three grain size ranges were produced using two stainless steel sieves with 63 µm and 180 µm mesh sizes. The medium range, i.e., 63-180 µm, was also used for the salinity experiments at ~36, ~28 and ~20." Did you measure the actual grain size distribution? I suggest making this measurement.

Thank you for pointing this out. This measurement has been attempted, but due to the high reactivity of the material with the matrix used for the analysis, no precise measurements were available. Therefore, a size fractionation using sieves was deemed more suitable.

Line 100 and throughout: How was omega determined?

$\Omega_A$ was determined using CO2SYS, and the measured TA, DIC, temperature and salinity at each data point, as mentioned in lines 153-156: "**$\Omega A$ and carbonate chemistry speciation were calculated from measured TA and DIC, providing temperature and salinity measurements, using CO2SYS (Sharp et al., 2021). To do so, the boric acid dissociation constant from Uppström (1974), the carbonic acid dissociation constant from Lueker et al. (2000), and the sulfuric acid dissociation constant of Dickson (1990) were selected.**"

Throughout: Why use free scale instead of total?

While total scale would have been more appropriate, the pH probe used throughout was measuring on the free scale. This was expected through the calibration using the mentioned buffers, and assessed and confirmed by providing measured TA, DIC, salinity and temperature data in CO2SYS and comparing the estimated $pH_F$ from the software with actual measured pH. After reviewing all data points, the estimated $pH_F$ from measured TA and DIC was identical to the measured pH with the above-mentioned probe.

Line 166: "After two hours, the maximum pHF recorded for the smaller grain size was 8.76 ±0.04, which continuously decreased to 8.68 ±0.00 between 11 and 12 hours after Mg(OH)2 addition. In contrast, the pHF for the medium and larger grain size increased to 8.72 ±0.00 and 8.68 ±0.03 after about eight hours and remained stable thereafter, respectively (Figure 1)." Why this behavior for small grains? no explanation is given.

Thank you for pointing this out. We indeed did not discuss this behaviour. We have added a paragraph in the discussion that covers this particular pattern (lines 338-346).

Line 184: "Finally, a small drop in DIC was observed after Mg(OH)2 addition in all experiments, of about 80, 30 and 140 µmol kg-1 in the small, medium and large grain size incubations, respectively." Why? What is the physical mechanism?

We believe that early $CaCO_3$ nucleation could have been responsible for such decrease. This is discussed in a newly added paragraph, lines 342-346.

Fig. 2: What accounts for the large spread in time among the 3 replicates?

Thank you for pointing this out, we believe that the reviewer refers to the different start of precipitation between replicates such as the small grain size experiments, with precipitation starting on days 3, 4 and 7. This is an interesting point that is hard to answer with certainty. We believe that the nucleation process happening between the addition of $Mg(OH)_2$ and the measured drop in TA may vary due to factors such as surface area and high pH in the diffusive boundary layer. It is possible

that the suitable conditions to start this nucleation process were met earlier on in one replicate compared to the other.

Line 200: "Starting pHF were slightly different, recorded at 7.99 ±0.05, 8.06 ±0.01 and 8.09 ±0.01 in the salinity 20, 28 and 36 incubations, and increased to a maximum of 9.19, ±0.00, 8.91 ±0.00 and 8.72 ±0.00, respectively. In all incubations, similar logarithmic trends were observed for pHF (Figure 3)." Maybe I missed it above. Do you add the same amount of Mg(OH)2 for all experiments (and if so what amount)? Or do you adjust the amount of Mg(OH)2 to target a certain starting pH or omega (and if so, why do you have slightly different starting pH / omega values?)

Thank you for the comment. For each salinity, different amounts of Mg(OH)$_2$ were added, (more being added at lower salinity) to increase $\Omega_A$ to similar values. These additions varied as at lower salinity, the concentration of calcium is lower (~5.7 mM instead of ~10 mM at salinity 35). Therefore, to increase $\Omega_A$ in lower salinity, a higher amount of alkalinity is required to increase $\Omega_A$ to the same level, here being the desired target of $\Omega_A$ ~9. The different starting pH$_F$ and $\Omega_A$ conditions are also the result of the lower salinity. NSW was diluted with MilliQ to decrease the salinity down to 28 and 20. Therefore, the starting pH$_F$ and $\Omega_A$ were lower. The text was edited as follow (lines 104-109): **"Varying amount of Mg(OH)2 were used in the salinity experiments. The decrease in dissolved [Ca] following dilution with MilliQ led to higher amounts of Mg(OH)2 to be added with decreasing salinity to reach a similar ΩA of about 9. Furthermore, preliminary tests conducted with the Mg(OH)2 powder used for these experiments, despite having reagent grade properties (>98% pure), have shown that only about 75% of the theoretical maximum TA was generated. Therefore, the Mg(OH)2 additions were adjusted accordingly, with additions varying from 23.3 mg kg-1 in the salinity 36 experiments (and all grain size experiments) to 30.2 mg kg-1 in the salinity 20 experiments."**

Fig. 3: From the caption to figure 3, the goal is same starting omega. Do you list somewhere how much Mg(OH)2 is added in each case?

The goal is the same $\Omega_A$ indeed. The amounts of Mg(OH)$_2$ added will be incorporated into the text (lines 107-109).

Line 219: "Similarly, following Mg(OH)2 additions, ΩA quickly increased to reach 8.6, 9.3 and 9.1 with increasing salinity." If targeting same omega, why do you have different values? What exactly determined how much Mg(OH)2 was added in each case? Somewhere the amount of Mg(OH)2 added in each case should be listed.

Thank you for the comment. As mentioned above, the Mg(OH)$_2$ additions were adjusted to take into account the lower [Ca] at lower salinity and the "missing alkalinity". Despite our best efforts, it is very likely that the 8.6 $\Omega_A$ recorded at salinity 20 is the result of early CaCO$_3$ precipitation before the observed TA drop. Such precipitation would have lowered the final TA and DIC values, resulting in a lower maximum $\Omega_A$.

Line 224: "Finally, DIC also decreased upon Mg(OH)2 additions. An initial DIC drop was observed directly after Mg(OH)2 additions of about 60 µmol kg-1 at the lowest salinity and 30 µmol kg-1 at the highest salinity. At a salinity of 28, a much smaller DIC drop was observed in one replicate. After a period of stable DIC conditions, DIC also dropped in a…" Why? What is the physical mechanism?

Thank you for highlighting this point, this "absence" of DIC drop is indeed questionable. While the drop in DIC in the salinity 36 and 20 experiments is clearly visible, the one in salinity 28 is more subtle. One could suspect that ingassing have decreased such DIC drop, but this cannot be verified in anyway. However, this observation has no implication of the remaining of the experiment, which is why this is not actively discussed.

Line 279: "Such ingassing is also occurring in the other experiments, but is likely hidden by the stronger pHF increase occurring during Mg(OH)2 dissolution." This doesn't make sense to me; I think you would see it just as well in the high pH case. Can you justify this statement quantitatively?

Thank you for the comment. We would like to clarify our point. When $Mg(OH)_2$ dissolves in seawater, the $pH_F$ increases due to the increase in TA. $CO_2$ will ingas as soon as the difference between water and atmospheric $pCO_2$ is negative (lower water $pCO_2$ lead to ingassing). If ingassing occurs, the increase in $CO_2$ will lead to a decrease in $pH_F$ (as per Bjerrum's plot). Therefore, these two processes are occurring simultaneously and in opposite direction. The smaller grain size reached a maximum $pH_F$ earlier than other grain sizes (~2h), suggesting full dissolution earlier, allowing for $pH_F$ to be recorded and the $CO_2$-driven decrease to be observed. In the other experiments, the slower dissolution rate over 8h hides the $CO_2$ ingassing impact until nearly the end of the $pH_F$ recorded data. The decrease in $pH_F$ and would only become visible late on (after 12h).

Line 281: "For salinity, we did not observe major differences in initial dissolution rates within the range of salinities tested, with dissolution rates for salinities 36, 28 and 20 estimated at 391.6 ±2.6, 359.8 ±0.2 and 301.9 ±0.3 µmol of TA mg-1 min-1…" These differences are quite significant: 25% variation!

This is a good point. The text has been edited in lines 293-296 to: **"For salinity, we did not observe major differences in there was a difference in initial dissolution rates within the range of salinities tested, with dissolution rates for salinities 36, 28 and 20 estimated at 391.6 ±2.6, 359.8 ±0.2 and 301.9 ±0.3 µmol of TA mg-1 min-1, respectively. While these differences are not as significant as those in the grain size experiments, the dissolution rate decreased by about 23% between salinity 36 and 20."**

Line 320: "Here, it could be higher pH levels and hence ΩA that are reached at a particle's surface as of having a larger diffusive boundary layer." Why is this true in a direction normal to the grain surface?

Thank you for the comment. One main outcome of the research is the potential two processes occurring simultaneously for $CaCO_3$ precipitation. The diffusive boundary layer plays a key role in providing high pH levels around the particles. These high levels allow for high $\Omega_A$ to be generated around the particles. Marion et al. (2009) reported that at salinity 35, an $\Omega_A$ of 12 was triggering pseudo-homogeneous precipitation. It is very likely that around the larger particles, these levels are much higher, allowing for homogeneous precipitation to occur within seconds, or even fractions of seconds. On the other hand, smaller particles present a smaller diffusive boundary layer, but the surface provided for the same amount of $Mg(OH)_2$ is much higher. There, the surface area allows for more $CaCO_3$ to nucleate, as suggested by Zhong and Mucci, 1989.

Line 381: "... while the second maintains a higher pH around larger particles due to a larger diffusive boundary layer compared to smaller particles, which increases precipitation rates." I think this claim needs to be supported with data and/or modeling.

Unfortunately, we cannot provide any modelling data, but we edited the text as follow (lines 408-412): **"One major finding of this research was that two processes seem to occur during CaCO3 precipitation in relation to grain size, one where the higher surface area of smaller particles could increases precipitation rates, while the second may maintains a higher pH around larger particles due to a larger diffusive boundary layer compared to smaller particles, which increaseds precipitation rates."**